# Terrestrial Laser Scanning to Predict Canopy Area Metrics, Water Storage Capacity, and Throughfall Redistribution in Small Trees

**Mariana D. Baptista** [1,*], **Stephen J. Livesley** [2], **Ebadat G. Parmehr** [3], **Melissa Neave** [1] and **Marco Amati** [1] 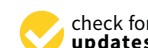

[1]   Centre for Urban Research, School of Global, Urban and Social Studies, RMIT University,
     Melbourne VIC 3001, Australia; melissa.neave@rmit.edu.au (M.N.); marco.amati@rmit.edu.au (M.A.)
[2]   School of Ecosystem and Forest Sciences, Faculty of Science, Burnley Campus, The University of Melbourne,
     Burnley VIC 3121, Australia; sjlive@unimelb.edu.au
[3]   Department of Geomatics, Faculty of Civil Engineering, Babol Noshirvani University of Technology,
     Babol 47148-71167, Iran; parmehr@nit.ac.ir
*   Correspondence: mariana.diasbaptista@rmit.edu.au; Tel.: +61-3-9925-9913

**Abstract:** Urban trees deliver many ecological services to the urban environment, including reduced runoff generation in storms. Trees intercept rainfall and store part of the water on leaves and branches, reducing the volume and velocity of water that reaches the soil. Moreover, trees modify the spatial distribution of rainwater under the canopy. However, measuring interception parameters is a complex task because it depends on many factors, including environmental conditions (rainfall intensity, wind speed, etc.) and tree characteristics (plant surface area, leaf and branch inclination angle, etc.). In the few last decades, remotely sensed data have been tested for retrieving tree metrics, but the use of this derived data for predicting interception parameters are still being developed. In this study, we measured the minimum water storage capacity ($C_{min}$) and throughfall under the canopies of 12 trees belonging to three different species. All trees had their plant surface metrics calculated: plant surface area (PSA), plant area index (PAI), and plant area density (PAD). Trees were scanned with a mobile terrestrial laser scan (TLS) to obtain their individual canopy point clouds. Point clouds were used to calculate canopy metrics (canopy projected area and volume) and TLS-derived surface metrics. Measured surface metrics were then correlated to derived TLS metrics, and the relationship between TLS data and interception parameters was tested. Additionally, TLS data was used in analyses of throughfall distribution on a sub-canopy scale. The significant correlation between the directly measured surface area and TLS-derived metrics validates the use of the remotely sensed data for predicting plant area metrics. Moreover, TLS-derived metrics showed a significant correlation with a water storage capacity parameter ($C_{min}$). The present study supports the use of TLS data as a tool for measuring tree metrics and ecosystem services such as $C_{min}$; however, more studies to understand how to apply remotely sensed data into ecological analyses in the urban environment must be encouraged.

**Keywords:** plant surface area; plant area index; plant area density; interception; runoff reduction; rainfall simulation

## 1. Introduction

Trees are an important component of the urban environment, as they can cool to moderate air temperatures [1–3], decrease air pollution [4,5], reduce noise [6], stimulate social connection [7], and reduce storm runoff effects [8,9], as well as many other benefits. Understanding, quantifying,

and communicating the benefits of trees is particularly important from an urban planning perspective, as a raised awareness of a scientific evidence base may help set policies and future management planning for urban forests.

Quantifying the role of urban trees in the mitigation of water runoff is important, as the frequency of floods has increased in densely urbanised areas in the last decades. The increase of impervious areas during urbanisation disrupts the natural cycling of water, reducing the number of naturally vegetated areas, and consequently, the permeability of the system. Additionally, stormwater infrastructure in many cities was designed for a less intensely urbanised landscape and is at, or exceeding, designed runoff capacity, and as a result, large rainfall events can frequently lead to flooding. Many cities in the world have set targets to increase tree canopy cover as one of the nature-based solutions to help mitigate the occurrence of floods [10–12].

However, predicting the impact of the urban forest on stormwater management is complex and so is its planning, primarily because predicting the impact of urban trees is not easy, given the complexity of trees' structural elements and the surrounding environment. Therefore, recent research has attempted to integrate knowledge from different disciplines to better predict tree metrics and their hydrological impact [13–15]. Part of this integration of knowledge involves reaching a precise understanding of the relationships between remotely sensed data and the effect of different canopies upon rainfall interception, storage, and throughfall.

Recently, terrestrial laser scanning (TLS) techniques have gained increased attention as a method to directly measure the 3D shape of tree canopies and consequently estimate different tree characteristics [16]. Although TLS data has been tested for retrieving tree metrics [17–22], the use of the derived TLS data for predicting ecosystem service processes, such as shading, pollution interception, or rainfall interception parameters, are still yet to be developed and tested.

Understanding how tree canopy characteristics at the whole-tree and leaf level influence canopy interception, storage, and throughfall redistribution dynamics is fundamental for advancing the use of trees for stormwater management. Both canopy water storage capacity and throughfall dynamics are driven by the combination of different tree attributes, as well as by rainfall characteristics and environmental conditions [23–25]. In the case of water storage capacity, plant surface area, roughness, and angle of inclination influence the volume of water that can be stored during and after rainfall ceases. Furthermore, woody parts of the tree intercept rainfall and drops from leaves, providing additional area for evaporation and drying. Fundamentally, trees with greater plant area, rougher surfaces, and lower inclination possess greater canopy water storage [26–29].

Similarly, the volume of water passing through the canopy (throughfall) is affected by tree configuration, which is a function of the arrangement of leaves and branches [30–33]. In addition, the spatial redistribution of throughfall under a tree canopy is modified and shaped by a variety of canopy characteristics. These changes have been studied, and importance has been given to the modification of rain characteristics, such as the kinetic energy of drops, drop size, and velocity [34,35]. Raindrop characteristics are linked to the occurrence of soil erosion and floods, mainly during high-intensity rainfall events [34]. Moreover, the availability of water and nutrients deposited with the throughfall flux positively influence soil biodiversity and root development [36–38]. However, few studies discuss how specific leaf and tree architectural traits, such as the arrangement of leaves and branches, influence water distribution under tree canopies [32,35,39,40].

This study proposes a new way to make use of TLS and thereby add value to the work of urban forestry professionals. It investigates the potential of deriving plant area metrics from TLS data clouds, so as to avoid expensive and laborious manual methods of leaf area data collection that require destructive sampling. It combines the use of remote sensing techniques with knowledge of water–tree interaction dynamics, aiming to validate the use of terrestrial laser scanning-derived data to predict both interception metrics and throughfall distribution.

In the first part of this study, we investigate the relationships between plant surface metrics and TLS-derived data. Based on previous works [17,19,20], the first hypothesis is that TLS-derived metrics

are good predictors for plant surface area metrics. Most of the previous approaches have used an algorithm to extract gap fractions from TLS point clouds, and from this, an approximation of leaf area index (LAI) [17,20,41,42]. Two approaches commonly applied have been the 3D voxel-based canopy profiling (VCP) method [41] and a 2D approach which converts "the point cloud data set from Cartesian coordinates to spherical coordinates in order to be similar to hemispherical photography" [20]. The present study differs from cited works by investigating whether the number of scanned points correlates with manually calculated plant area metrics. TLS-derived metrics are then tested to predict one of the water storage capacity parameters, the minimum storage capacity ($C_{min}$). The second hypothesis is that TLS-derived metrics are effective in predicting both $C_{min}$ and throughfall redistribution.

In the second part of this paper, the process of spatial redistribution of throughfall is studied in more detail, aiming to understand how tree structure can affect the total throughfall redistribution. Therefore, the TLS data will be used in a subcanopy analysis, providing a novel approach [33,39].

The specific objective of this study is to investigate: (a) the efficacy of TLS data to predict urban tree morphological characteristics by correlating directly measured plant area metrics with TLS-derived metrics, (b) the use of TLS-derived metrics to predict water storage capacity, and (c) whether the use of TLS-derived metrics can provide information at the subcanopy scale to understand canopy throughfall distribution. Results may add to previous knowledge on the use of TLS data for canopy metric estimation, and may possibly be used in a way to predict interception parameters and throughfall distribution. Also, the results will improve the understanding of the throughfall process, providing a theory-based discussion for future research.

## 2. Methodology

Rainfall was simulated above trees (at 2.7 m height) in pots of three different species in a controlled indoor environment. Trees were scanned by a TLS to generate the point clouds to measure canopy projected area and volume and derive the following metrics: number of points (NP), number of points per canopy projected area (NPA), and number of points per canopy volume (NPV). Trees were then destructively sampled, and plant surface area (leaf area plus branch area) was measured. Then, TLS-derived data was correlated to the minimum storage capacity ($C_{min}$) and throughfall collected under individual tree canopies during simulated rainfall.

### 2.1. Trees

Canopy water storage and throughfall were measured for 12 trees of three different species: *Ulmus procera*, *Platanus × acerifolia*, and *Corymbia maculata*. These species were selected for their distinct canopy characteristics and because they are commonly used in the streetscapes of the City of Melbourne. The studied trees were grown in 100-litre pots and had comparable basal stem diameters ranging from 6.2 to 9.7 cm, and initial canopy volumes that ranged from 5.61 m$^3$ to 7.99 m$^3$ (Table 1). For the remainder of this paper, the individual *Corymbia* trees are referred to as CM1 to CM4, the *Ulmus* trees are referred to as UP1 to UP4, and the *Platanus* trees are referred to as PA1 to PA4.

### 2.2. Canopy Surface Area Manipulation and Measurement

The leaf density was manipulated with the purpose of simulating differences in water storage capacity from a canopy in good to poor health. Leaf surface area was manipulated in stages: after canopy water storage measurements and TLS scanning was completed on trees with 100% of their canopy (labelled as 'Full canopy'), every other leaf on a branch was removed and leaf areas were directly measured using a leaf area meter (LI-3100 Area Meter, Li-cor, Lincoln, USA). Leaves were then oven-dried at 60 °C for 48 h to determine leaf dry mass, and from this, a ratio of leaf area to dry leaf mass was established. The remaining tree canopies had a density of approximately 50% (or 'Half canopy'). After canopy water storage measurements and TLS scanning, the trees again had half of their remaining leaves removed, resulting in a canopy density of approximately 25%

(or 'Quarter canopy'). These leaves were oven-dried and their surface area estimated from the ratio of leaf area to dry leaf mass. The canopy water storage measurements and TLS scanning were then performed on these quarter canopies, and then finally, all remaining leaves were removed and oven-dried, and then their leaf area was calculated for each canopy density treatment [43]. The final leaf removal process resulted in a canopy without leaves (or 'Woody').

After water storage capacity measurements and TLS scanning, the woody material for each tree was collected and divided into two groups depending on their diameter class: ≥1 cm or <1 cm. Each branch group had its length, area, and volume estimated [43].

## 2.3. Rainfall Simulation

This study uses an indoor rainfall simulator (Figure 1). Full details for this method and for keeping constant precipitation rates, uniformity, and raindrop size are available in Refs. [43,44].

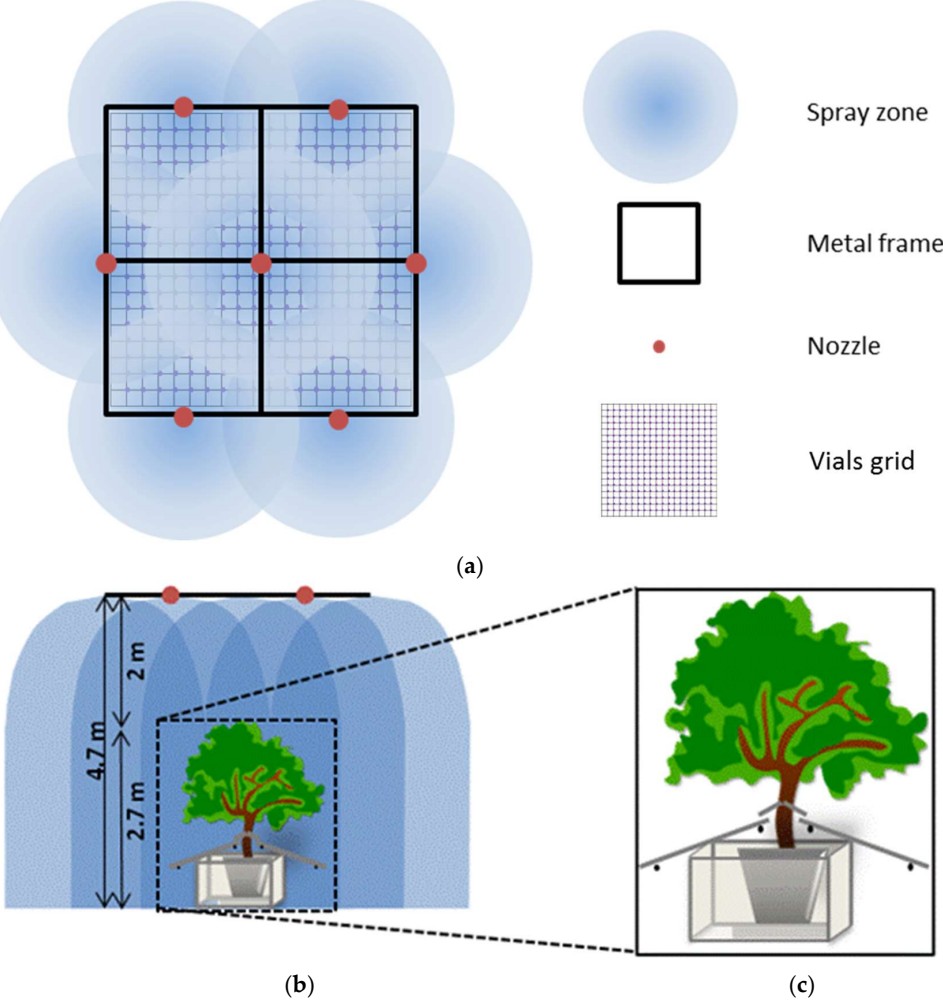

**Figure 1.** Top view of simulator spray zone with vials grid (**a**); Rainfall simulator cross-section view (**b**); small plastic roof attached to the tree and plastic sheets to prevent water from dripping into the pot (**c**).

Rainfall intensity was set at 2.5 mm/h. A rainfall rate of 2.5 mm/h for 15 min is classified as being very frequent in Melbourne, according to the Intensity-Frequency-Duration design tool [45].

All trees were top-trimmed to ensure they did not exceed a height of 2.7 m, so that rainfall distribution was uniform once it intercepted the top of a tree canopy. Many street trees in Melbourne and other cities that retain overhead power lines are commonly lopped to maintain a clearance zone between the top of the tree and the power lines. Our experimental setup, therefore, replicates real-world

conditions. Full details on how the rainfall simulator was constructed, trialled, and operated with trees beneath are presented in Ref. [43].

*2.4. Canopy Water Storage Measurements*

Each tree was placed on a balance (150 kg capacity, 20 g resolution, EM-150KAL, A&D Weighing, Thebarton, Australia) to continuously measure the change in mass before, during, and after a 15-min simulated rainfall event (0.6 mm). From this, the maximum and minimum canopy water storage were calculated from the changes in mass balance (i) during the 15 min of rainfall ($C_{max}$) and (ii) from the end point of the rainfall event followed by 15 min of excess water dripping from the canopy ($C_{min}$). All trees had the canopy water storage measured for the different canopy density treatments: Full canopy, Half, Quarter, and Woody. Full details on these calculations and the experimental approach used to collect the data are presented in Ref. [43].

*2.5. Throughfall Distribution*

During measurements of water storage capacity for trees in their full (100%) canopy, total throughfall was measured using small graduated plastic vials (50 mL, opening diameter 2.8 cm, model PP, Sarstedt Inc., Germany). Vials were evenly distributed under the canopy in an 11-cm spaced grid of 18 × 17, with 9 vials being excluded to fit the tree stem (n = 297). These vials were held in place by plastic trays attached to tables to ensure they were truly vertical in orientation and in exactly the same position under each rainfall simulation (Figure 2). Vials were labelled according to their position in the grid and remained in position during the 15 min of rainfall simulation followed by 15 min of post-rainfall dripping. After these 30 min, vials were weighed on a 2-decimal place balance (3100 g capacity, 0.01 g resolution, GF3000, A&D Weighing, Thebarton, Australia). Any water on the outside surface of a vial was wiped away before weighing. The average mass of a plastic vial was subtracted from the total mass and the remaining mass converted to water volume (mL), assuming the density of water as 1.0 g/mL. Each water volume was then converted to water throughfall (mm) by dividing the mass by the area of the opening of the vial (6.16 cm$^2$). The vials covered about 5% of the total area under each tree canopy.

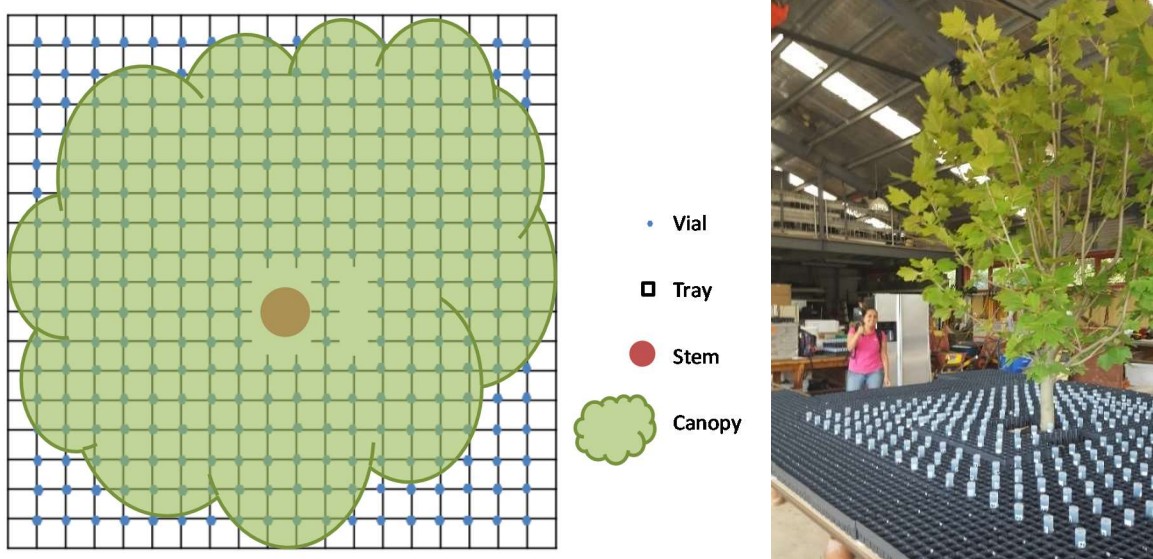

**Figure 2.** The arrangement of vials under the canopy: plan view (**left**) and photo (**right**).

### 2.6. TLS Data Collection and Processing

A three-dimensional image of each tree canopy was captured using a hand-held laser scanner (ZEB1, GeoSLAM Ltd., Nottingham, UK). All trees were scanned for each one of the canopy density treatments: Full canopy, half, quarter, and woody, resulting in a total of 12 point clouds for each species (three replicate trees were each measured four times at 100%, 50%, 25%, and 0% canopy density). Each tree was scanned by walking around them three times at a distance of 1–2 m from the tree with the ZEB1 laser scanner held out in front to maximise point cloud accuracy and density. Studies using TLS data to retrieve tree metrics have recommended the use of multiple scans to reduce the occlusion of objects [13,46]. To standardise the point clouds, duplicated points should be removed from the dataset. Therefore, the random downsampling method was applied, and the two or more points that were within 2 mm of another one would be retained, and the others removed. This process detected 1% of points as being duplicated points. Then, a statistical denoising algorithm was applied to remove outlier points, also called dongle points, from the laser point cloud. Based on a 3-sigma test, points with a confidence level of 99.7% were considered as inliers. To apply this method, the mean and standard deviation of 10 neighbour points were calculated, and points which did not pass the 3-sigma test were detected as noise points. In other words, points with a weak connection to sample points were detected as noise points and were removed from the point cloud. In this research work, less than 2% of points were detected as outliers.

Point clouds data were processed using CloudCompare 2.6.2 software. A concave hull method was then used to calculate tree canopy metrics, such as canopy project area and canopy volume, from the point cloud. A 2D graph of point density distribution was created in Cloud Compare from the processed point cloud. Firstly, the 3D point cloud was converted to two-dimensional data by ignoring z-axis information. Then, the density of points was calculated based on the number of neighbours in a 10-cm pixel. The point density map was created and converted to a matrix, where the values of point density were scaled up to a grid composed of $11 \times 11$-cm pixels. For throughfall analyses, the 3D point cloud was converted into a grid-based 2D graph according to the density of points from the top view of the whole tree and then scaled to the same resolution as the throughfall grid data collection. Point cloud density was classified into 5 different percentiles: low, low to medium, medium, medium to high, and high density. These classes of density were correlated with throughfall categories for each pixel.

### 2.7. Data Presentation and Statistical Analysis

Metrics derived from TLS data were presented as the number of points (NP); the number of points per area (NPA), which represents NP divided by the calculated canopy projected area; and the number of points per volume (NPV), which represents NP divided by the estimated concave canopy volume. TLS-derived metrics were correlated with the directly measured plant area: linear regressions analyses were performed between plant surface area (PSA) and number of points (NP); plant area index (PAI) and number of points per area (NPA); and plant area volume (PAV) and number of points per volume (NPV). Analyses of covariance (ANCOVA) were performed to identify differences between species, with plant surface metrics (PSA, PAI, and PAD) held as the dependent variable and TLS-derived metrics (NP, NPA, and NPV) as the covariates. These analyses are intended to test the hypothesis that TLS-derived metrics are good predictors for plant surface metrics.

In the second part of analyses, TLS-derived metrics were correlated to the interception parameter, $C_{min}$. Analyses of covariance (ANCOVA) were performed to identify differences between species, with $C_{min}$ held as the dependent variable and TLS metrics (NP, NPA, and NPV) as the covariates. These analyses intended to answer if TLS-derived metrics are effective in predicting $C_{min}$.

For the throughfall redistribution analyses, the average throughfall (mm), the coefficient of variation, and the standard deviation were calculated for every simulation. The average of throughfall for each tree was calculated based on the volume of water collected in vials that were covered by the canopy. For this reason, the point density matrix and data of throughfall volume collected in the vials

which overlapped and were outside (beyond) the canopy cover area were excluded for this calculation. During this experiment, two trees needed to be replaced, UP2 and UP3. In this process, UP3 was not scanned for the full (100%) canopy condition; therefore, throughfall calculations for this tree were based on the total incident area. A Kendall correlation test was performed to analyse the correlation between the throughfall volume and density of points.

To analyse throughfall distribution, each point of collection was categorised according to the interception rate calculated. To do so, the interception for each point was calculated by subtracting the collected throughfall values from the average incident rainfall, obtained from 4 control treatments without trees. Then, each point was categorised: negative results for throughfall were considered as "concentration" zones, and positive results were considered as "reduction" zones. This information was correlated with the 5 different classes of density derived from the TLS point clouds. Maps of throughfall distribution, throughfall categories, and class of canopy density were created on ArcGIS, and statistical analyses were performed on RStudio.

## 3. Results

### 3.1. Correlations between TLS Data and Tree Metrics

Firstly, the number of points for each tree was assessed and derived tree metrics were calculated from the scanned data (Table 1). TLS metrics were correlated to directly measured metrics to validate the effectiveness of TLS data in predicting plant surface metrics (Figure 3).

**Table 1.** Values of plant surface metrics and laser scanner metrics and interception parameters collected for each studied tree.

| | PSA ($m^2$) | PAI ($m^2/m^2$) | PAD ($m^2/m^3$) | NP (points) | NPA (points/$m^2$) | NPV (points/$m^3$) | $C_{max}$ (mm) | $C_{min}$ (mm) | Throughfall (mm) |
|---|---|---|---|---|---|---|---|---|---|
| CM 1 | 4.9 | 2.1 | 1.0 | 68,365 | 29,468 | 13,856 | 0.25 | 0.09 | 4.53 |
| CM 2 | 3.9 | 1.6 | 0.8 | 60,732 | 24,293 | 12,014 | 0.18 | ** | 4.67 |
| CM 3 | 5.6 | 2.0 | 0.8 | 91,938 | 32,718 | 13,182 | 0.19 | 0.08 | 5.19 |
| CM 4 | 5.4 | 1.9 | 1.0 | 75,053 | 26,427 | 13,667 | 0.18 | 0.08 | 4.18 |
| PA 1 | 9.5 | 3.7 | 1.3 | 91,518 | 35,472 | 12,159 | 0.52 | 0.31 | 3.49 |
| PA 2 | 9.2 | 3.1 | 1.0 | 80,686 | 26,895 | 9004 | 0.40 | 0.23 | 4.08 |
| PA 3 | 11.4 | 4.0 | 1.6 | 81,179 | 28,787 | 11,523 | 0.57 | 0.32 | 3.85 |
| PA 4 | 10.0 | 3.2 | 1.2 | 96,613 | 30,477 | 11,475 | 0.44 | 0.29 | 3.95 |
| UP 1 | 8.2 | 5.1 | 1.1 | 55,945 | 34,966 | 7409 | 0.71 | 0.31 | 4.48 |
| UP 2 | 8.6 | 5.8 | 1.3 | 51,684 | 34,687 | 7868 | 0.83 | 0.39 | 4.53 |
| UP 3 | 7.9 | 5.3 | 1.2 | 41,920 | 27,947 | 6431 | 0.77 | 0.41 | 4.50 * |
| UP 4 | 6.8 | 4.5 | 1.1 | 37216 | 24811 | 5864 | 0.67 | 0.39 | 4.27 |

\* Including total incident area; \*\* Value excluded due to measurements error; PSA: Plant surface area; PAI: Plant area index; PAD: Plant area density; NP: Number of points; NPA: Number of points per canopy area; NPV: Number of points per canopy colume; Cmax: maximum storage capacity; Cmin: minimum storage capacity.

Linear regressions between TLS and manually measured metrics showed a significant relationship for all species ($R^2 > 0.7$). The variance between the variable related to scanned points and those related to tree measurements were not significantly different for any of the regressions performed, as validated by the F-test (Table 2). All regressions for *U. procera* presented higher values for coefficient *b*, showing steeper slope compared to the other two species (Figure 3).

ANCOVA tests for NP and PSA, and NPV and PAD showed a significant difference between the studied species ($p$-value < 0.05). On the other hand, the relationship between the NPA and PAI showed no significant difference between species ($p$-value > 0.05).

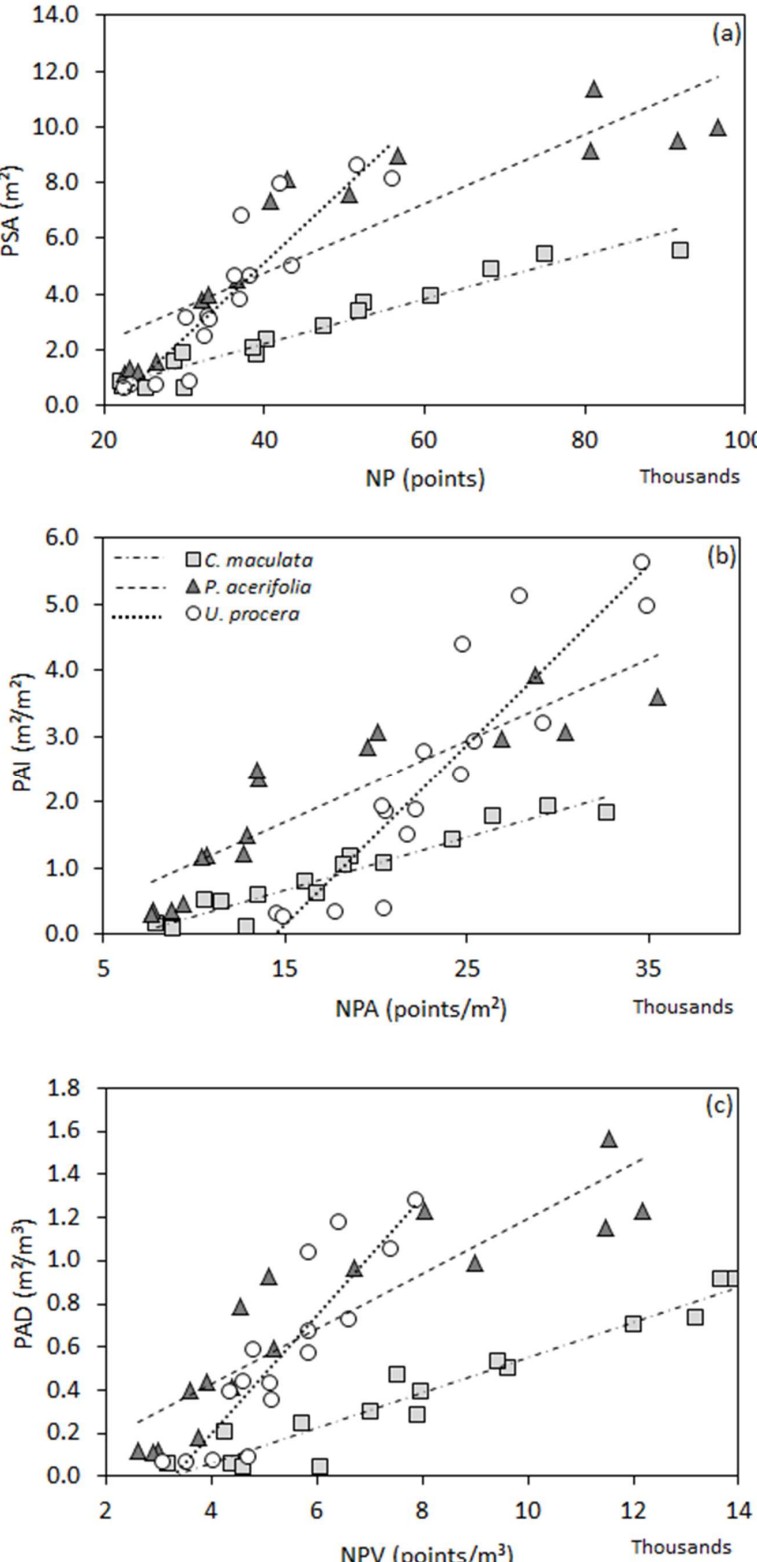

**Figure 3.** Linear regressions between TLS-derived metrics and plant surface metrics. (**a**) Number of points (NP) × plant surface area (PSA); (**b**) Number of points per canopy area (NPA) × plant area index (PAI); (**c**) Number of points per canopy volume (NPV) × plant area density (PAD).

**Table 2.** Summary of linear regressions parameters and significance tests for correlations between scanning-derived and plant surface metrics.

| Species | x | y | a | b | $R^2$ | F-Test |
|---|---|---|---|---|---|---|
| *U. procera* | NP | PSA | −5.677 | $3 \times 10^{-4}$ | 0.826 | $8.3 \times 10^{-50}$ |
| *P. acerifolia* | NP | PSA | −0.182 | $1 \times 10^{-4}$ | 0.792 | $1.0 \times 10^{-54}$ |
| *C. maculata* | NP | PSA | −1.006 | $1 \times 10^{-4}$ | 0.942 | $4.0 \times 10^{-58}$ |
| *U. procera* | NPA | PAI | −3.944 | $3 \times 10^{-4}$ | 0.823 | $9.9 \times 10^{-50}$ |
| *P. acerifolia* | NPA | PAI | −0.156 | $1 \times 10^{-4}$ | 0.802 | $8.9 \times 10^{-55}$ |
| *C. maculata* | NPA | PAI | −0.540 | $8 \times 10^{-5}$ | 0.927 | $4.4 \times 10^{-58}$ |
| *U. procera* | NPV | PAD | −0.900 | $3 \times 10^{-4}$ | 0.811 | $1.2 \times 10^{-49}$ |
| *P. acerifolia* | NPV | PAD | −0.077 | $1 \times 10^{-4}$ | 0.813 | $1.2 \times 10^{-54}$ |
| *C. maculata* | NPV | PAD | −0.267 | $8 \times 10^{-5}$ | 0.927 | $5.9 \times 10^{-58}$ |

Notes: x: independent variable; y: dependent variable; a: intercept; b: slope; $R^2$: coefficient of determination; F-test at 99% of significance.

### 3.2. Correlation between TLS Data and Rainfall Interception Parameters

Regression tests were then performed to assess the effectiveness of using TLS-derived metrics in the prediction of water storage capacity of the studied tree species. Correlations between measured and TLS metrics were significant for all correlations, except for NPA $\times$ $C_{min}$.

The ANCOVA analyses returned several significant correlations between the scanned metrics and $C_{min}$. ANCOVA tests demonstrated that all interactions were significantly different between species and number of points ($p < 0.05$). Therefore, linear models were separately performed for each species (Table 3, Figure 4).

For all linear regressions, *U. procera* is the species showing greater water storage capacity when compared to *P. acerifolia* and *C. maculata*, even if the number of points scanned is lower than for the other two species. However, *U. procera* presented the lowest coefficient of determination ($R^2 < 0.6$) compared to the other two species, highlighting that this relationship was more subject to random effects than the other trees.

**Table 3.** Summary of linear regressions parameters for correlations between scanning-derived metrics and $C_{min}$.

| Species | x | y | a | b | $R^2$ |
|---|---|---|---|---|---|
| *U. procera* | NP | $C_{min}$ | $1 \times 10^{-5}$ | −0.161 | 0.512 |
| *P. acerifolia* | NP | $C_{min}$ | $4 \times 10^{-6}$ | −0.063 | 0.929 |
| *C. maculata* | NP | $C_{min}$ | $8 \times 10^{-7}$ | 0.019 | 0.667 |
| *U. procera* | NPA | $C_{min}$ | $2 \times 10^{-5}$ | −0.018 | 0.544 |
| *P. acerifolia* | NPA | $C_{min}$ | $1 \times 10^{-5}$ | −0.062 | 0.951 |
| *C. maculata* | NPA | $C_{min}$ | $2 \times 10^{-5}$ | −0.130 | 0.634 |
| *U. procera* | NPV | $C_{min}$ | $7 \times 10^{-5}$ | −0.194 | 0.580 |
| *P. acerifolia* | NPV | $C_{min}$ | $3 \times 10^{-5}$ | −0.060 | 0.943 |
| *C. maculata* | NPV | $C_{min}$ | $5 \times 10^{-6}$ | 0.016 | 0.700 |

Notes: x: independent variable; y: dependent variable; a: intercept; b: slope; $R^2$: coefficient of determination.

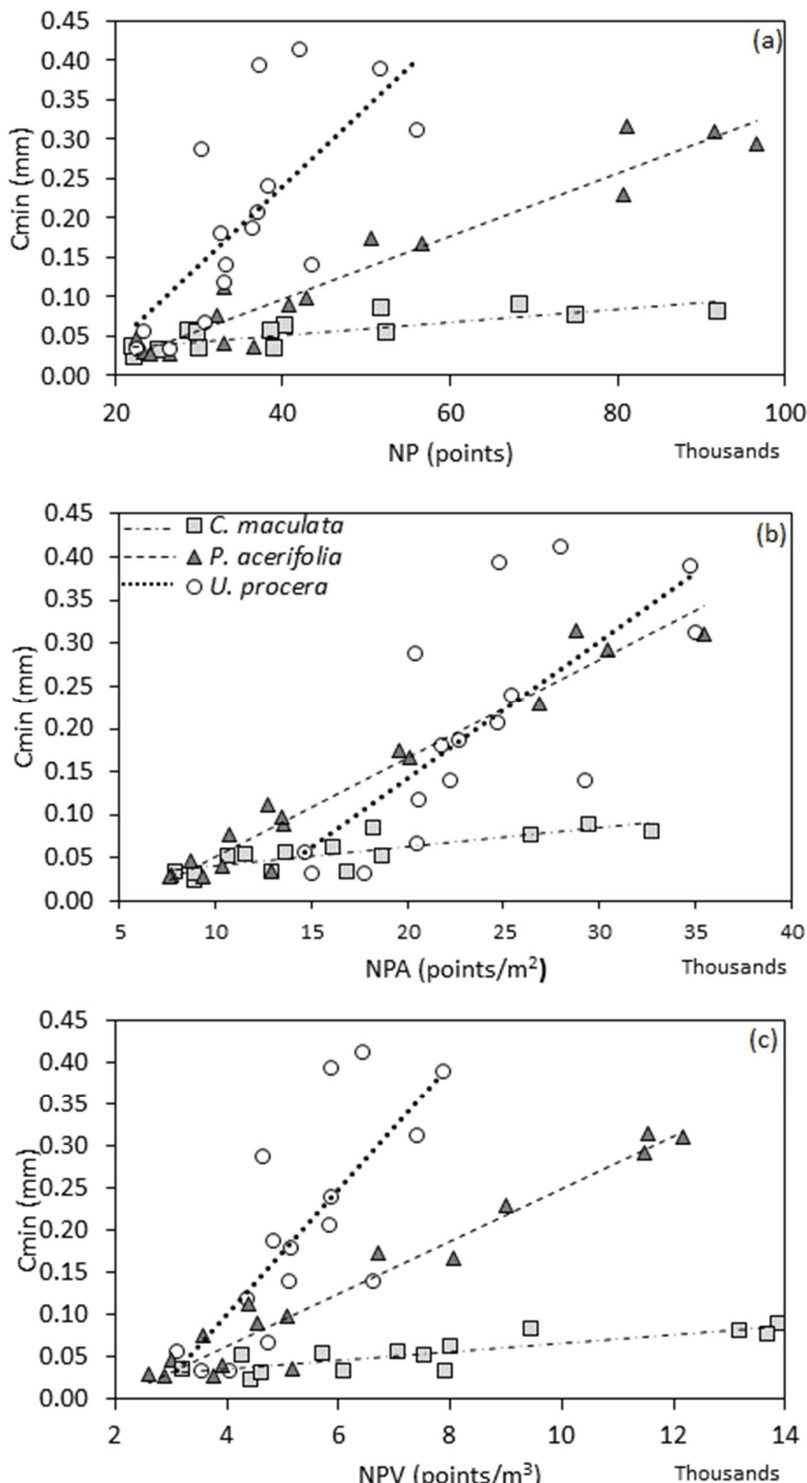

**Figure 4.** Linear regression between TLS-derived metrics (**a**) Number of points: NP; (**b**) Number of points per canopy projected area: NPA; (**c**) Number of points per volume: NPV; and water storage capacity parameter ($C_{min}$).

### 3.3. Throughfall and Spatial Redistribution

An average rainfall of 0.9 mm was calculated from four controlled treatments (no trees), with a standard deviation of 0.3 and a coefficient of variation of 28% (Table 4). The average values of throughfall per species were 1 mm, 0.8 mm, and 0.9 mm for *C. maculata*, *P. acerifolia*, and *U. procera*, respectively (Table 4).

**Table 4.** Average rainfall (control treatment) and throughfall in mm (Avg), standard deviation (Std), and coefficient of variation (CV) for studied trees.

| | **Control** | | | *C. maculata* | | | *P. acerifolia* | | | *U. procera* | | |
|---|---|---|---|---|---|---|---|---|---|---|---|---|
| ID | Avg | Std | CV | Avg | Std | CV | Avg | Std | CV | Avg | Std | CV |
| 1 | 1.0 | 0.3 | 30% | 0.9 | 0.3 | 38% | 0.8 | 0.8 | 107% | 0.9 | 0.5 | 54% |
| 2 | 0.9 | 0.2 | 28% | 1.0 | 0.5 | 49% | 0.8 | 0.9 | 112% | 0.9 | 0.5 | 48% |
| 3 | 0.8 | 0.3 | 27% | 1.0 | 0.7 | 68% | 0.8 | 0.8 | 100% | 0.9 | 0.4 | 42% |
| 4 | 0.8 | 0.3 | 28% | 0.9 | 0.4 | 49% | 0.8 | 0.7 | 84% | 0.9 | 0.5 | 55% |

In the subcanopy analysis, a slight similarity of the canopy contour may be noticed in the throughfall distribution map when overlain by the canopy density map, showing the greater value of throughfall in areas around the canopy's dripping edge (Figures 5–7). During this experiment, two trees needed to be replaced (UP2 and UP3), and unfortunately, UP3 was not scanned and not counted in this throughfall analysis.

Kendall's correlation presented a *p*-value = $3.793 \times 10^{-8}$ and coefficient tau = $-0.0769$, indicating a significant negative correlation between canopy density—represented by the density of points—and throughfall, but a weak coefficient of correlation.

For this reason, density was grouped into five different classes and throughfall data was categorised according to the interception calculation for each point where throughfall was collected (Figure 8). Points categorised as concentration indicate a throughfall volume higher than the incident precipitation (negative interception), whereas reduction points indicate a throughfall volume lower than the incident precipitation (positive interception).

For all species, the majority of canopies were classified as low-density zones. High-density zones were visually identified in the scanned data as predominantly woody (Table 5). According to the categorical classification for the throughfall collection points, *C. maculata* and *P. acerifolia* presented more reduction zones than concentration ones; on the other hand, *U. procera* presented a slightly higher number of concentration zones than reduction ones.

**Table 5.** Distribution of reduction and concentration of throughfall zones according to class of point density.

| | | *U. procera* (n = 3) | | *P. acerifolia* (n = 4) | | *C. maculata* (n = 4) | |
|---|---|---|---|---|---|---|---|
| **Density Classes** | **Percentile** | C * | R ** | C | R | C | R |
| Low | <20% | 162 | 98 | 214 | 403 | 276 | 340 |
| Low to medium | 21–40% | 41 | 57 | 69 | 170 | 81 | 100 |
| Medium | 41–60% | 22 | 34 | 17 | 38 | 29 | 36 |
| Medium to high | 61–80% | 9 | 26 | 1 | 15 | 6 | 11 |
| High | >80% | 3 | 6 | 2 | 3 | 1 | 1 |
| Subtotal | | 237 | 221 | 303 | 629 | 393 | 488 |
| Total | | 458 | | 932 | | 881 | |

\* Concentration; \*\* Reduction.

Despite the weak correlation between throughfall volume and point density, Mann–Whitney U-test analysis showed a significant association between density and throughfall categories for *P. acerifolia* (*p* = 0.009, *p* < 0.05) and *U. procera* (*p* = 0.000, *p* < 0.05), whilst the difference was not significant for *C. maculata* (*p* = 0.302, *p* > 0.05) (Figure 9). As can be expected, the denser the canopy, as measured by the TLS, the more the canopy acts as a shelter.

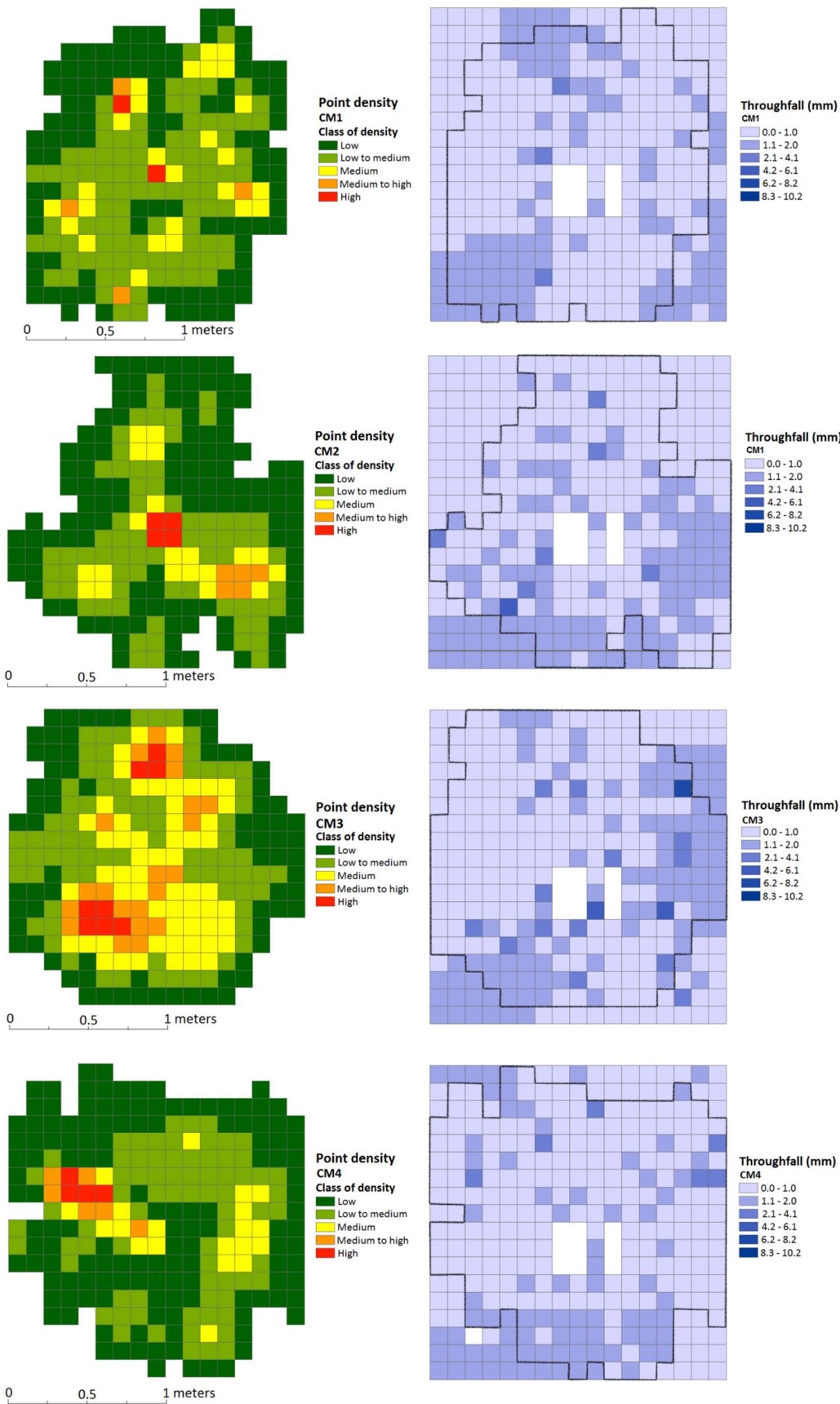

**Figure 5.** Canopy density and throughfall redistribution maps for all studied *C. maculata* trees.

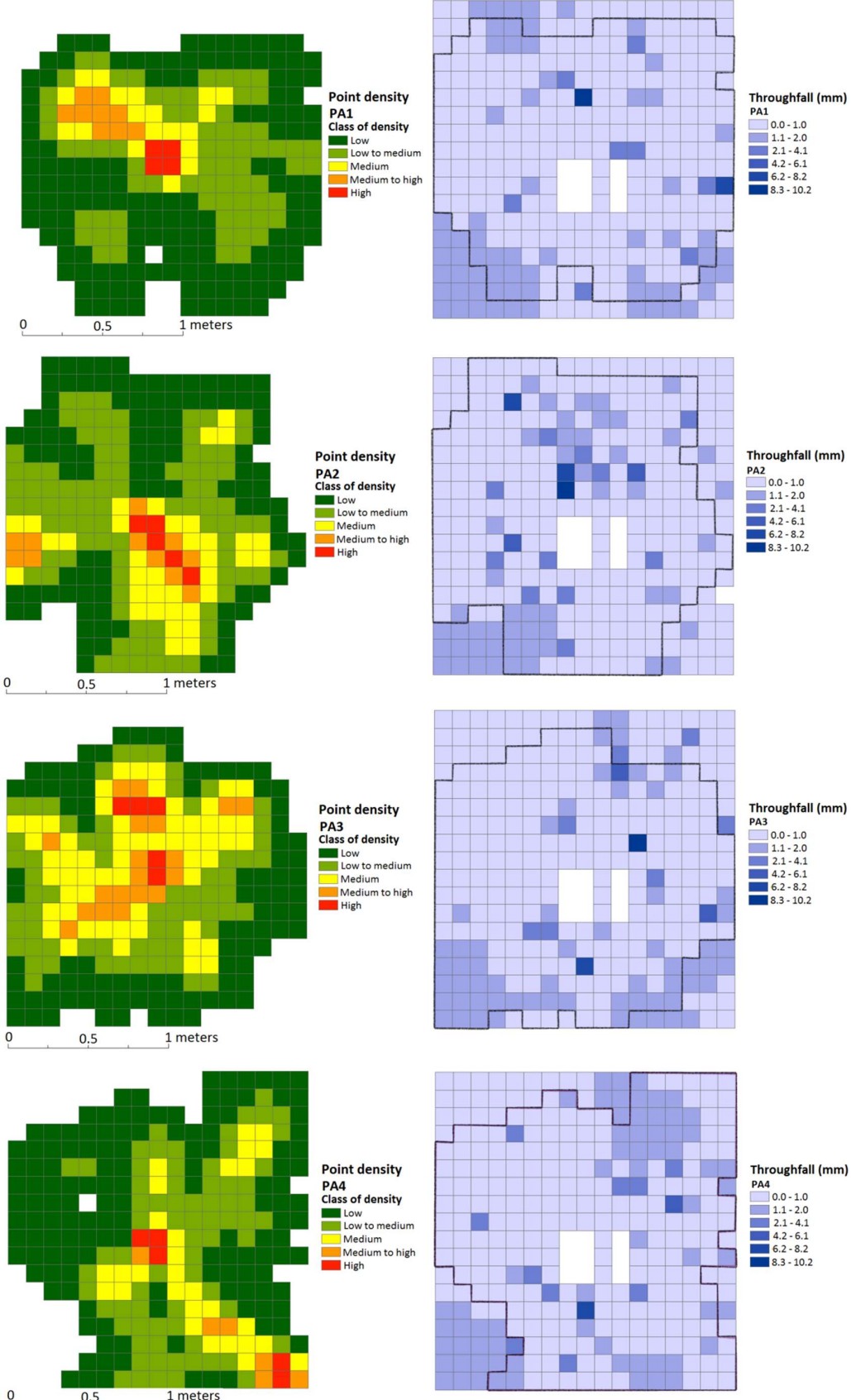

**Figure 6.** Canopy density and throughfall redistribution maps for all studied *P. acerifolia* trees.

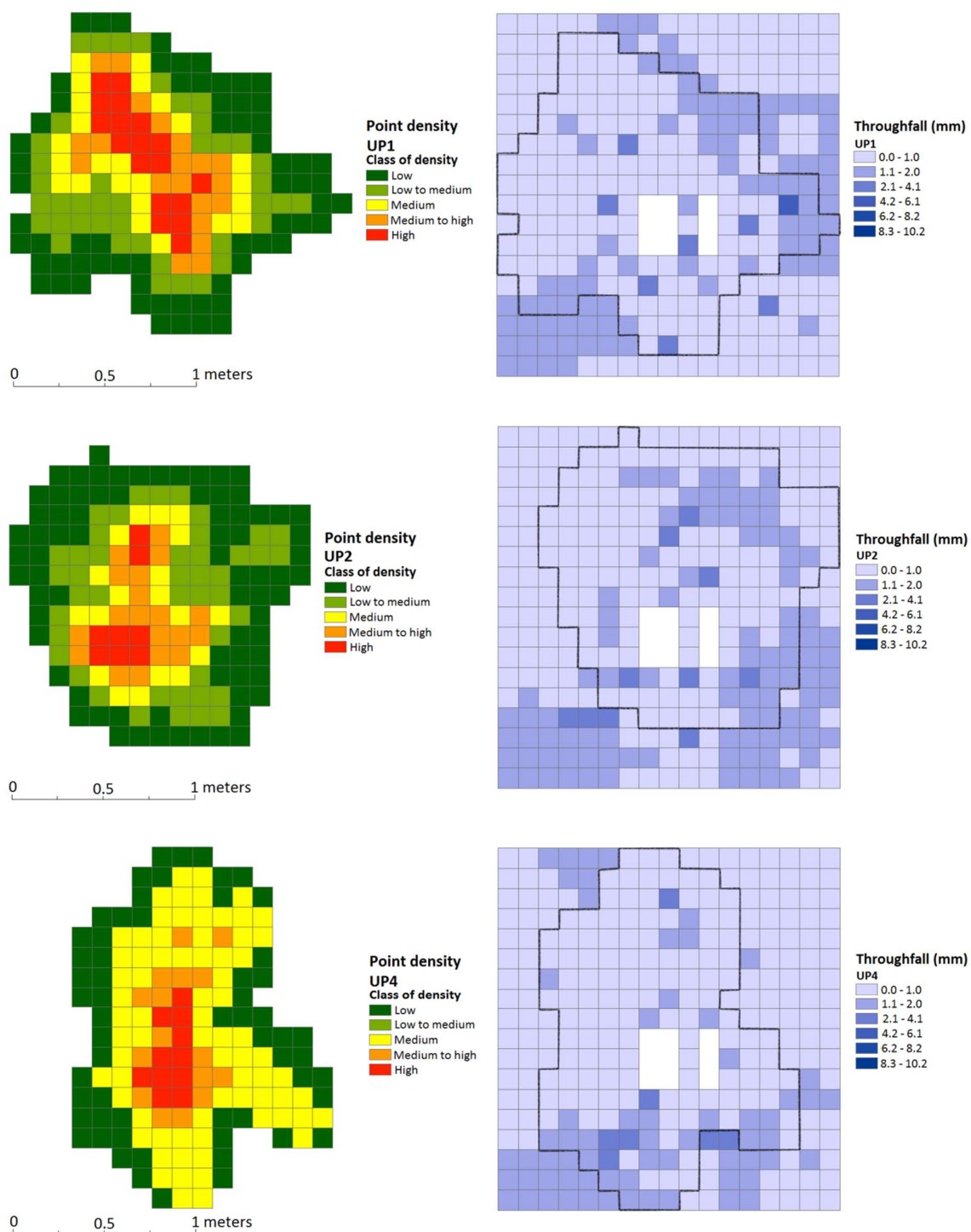

**Figure 7.** Canopy density and throughfall redistribution maps for three studied *U. procera* trees.

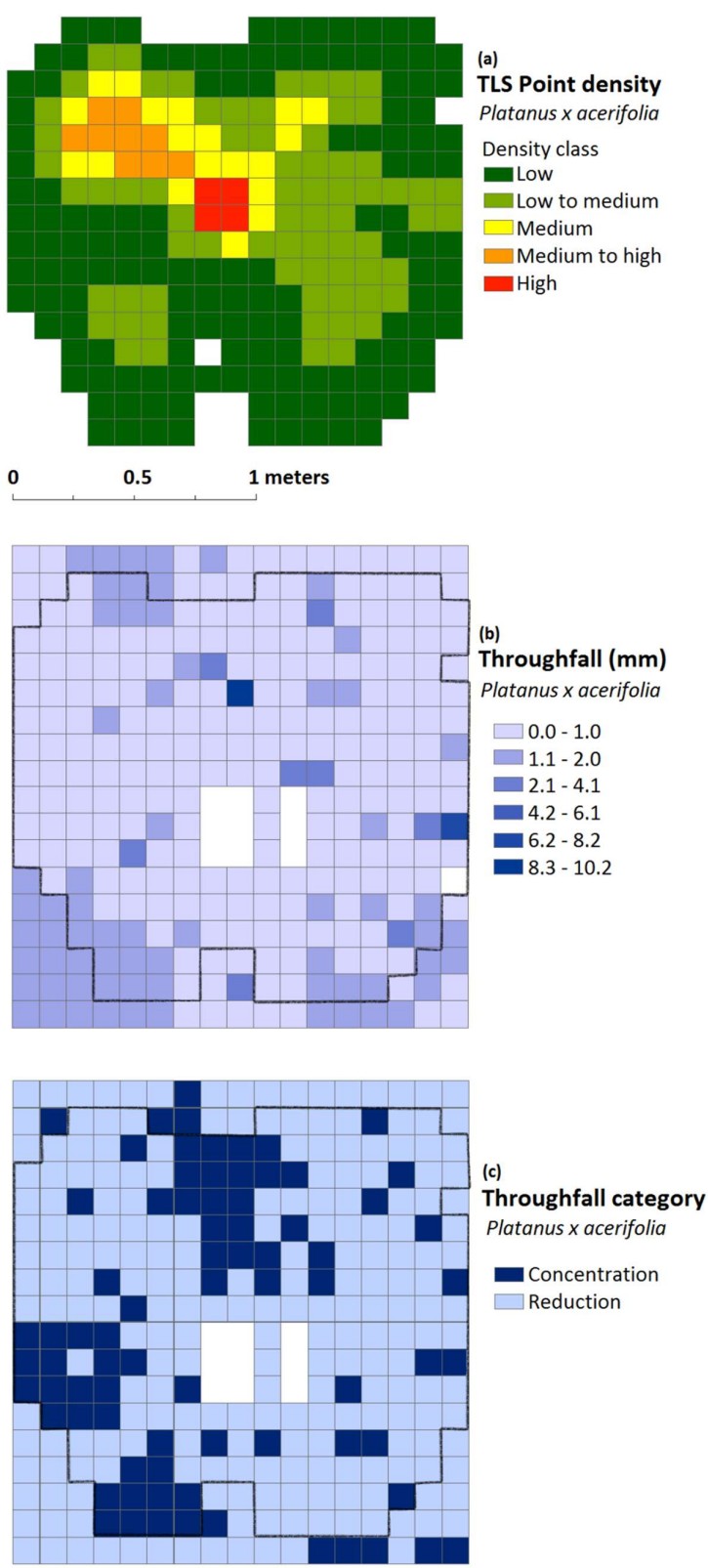

**Figure 8.** (**a**) Point density map; (**b**) throughfall distribution; and (**c**) throughfall category map showing zones of reduction and concentration for tree PA1.

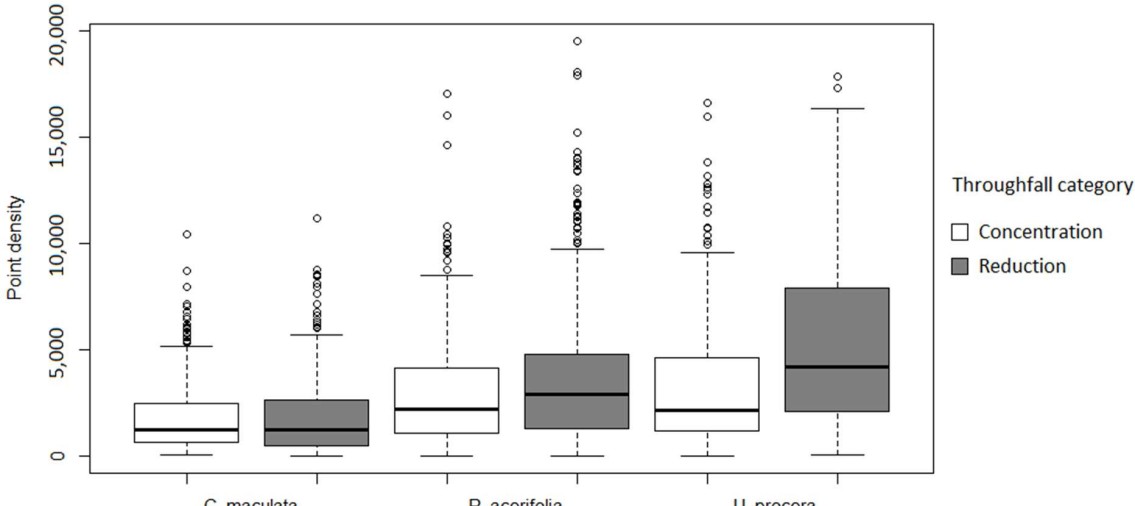

**Figure 9.** Box plots of the distribution of point density into the two different throughfall categories for each species.

## 4. Discussion

### 4.1. Correlation TLS Data and Tree Metrics

For all combinations tested, linear regressions showed a strong correlation between the measured and scanned metrics (Table 2, $R^2 > 0.70$; *p*-value < 0.01). Likewise, studies retrieving LAI from TLS data have shown a significant correlation between plant surface metrics and remotely sensed data [17,20,22,46]. This study confirms the strong correlation between surface area metrics and the number of scanned points, and therefore provides encouragement for the use of TLS data for tree metrics' estimation, although the results show that there is variation in this between species.

Significant differences between the species in ANCOVA analysis indicate that differences in plant structure, such as leaf size and angle, have an influence on a different number of points per canopy. Because each point in the laser scanning data corresponds to laser beams that reached any canopy surface (branch or leaf), we expected that the number of scanned points would correlate with actual values and not suffer any species influence if the plant structures were in the same level of scanning. For example, *U. procera* presented a greater slope in all regression analyses when compared to the other two species. In this case, the greater slope is an indicator that the results for plant surface metrics were greater in comparison with the scanned metrics. However, *U. procera* presented the lowest coefficient of determination, indicating a weaker correlation between plant surface metrics and scanned metrics. On the other hand, *C. maculata* presented smaller results for plant surface metrics in relation to scanned metrics and a higher coefficient of determination compared to *U. procera*. Therefore, this suggests that the number of points may be underrepresented for *U. procera*. Because the accuracy of the laser scanner is about 0.02 m, smaller leaves are harder to capture with the device [19]. In this case, *U. procera* leaf size may be affecting results, as smaller leaves tend to return a less accurate representation of leaf surface area. On the other hand, the size of leaves may favour the retrieving of the leaf surface area for *C. maculata* and *P. acerifolia*.

The angle of leaves has also an influence in the accuracy of data acquired [18,47]. Smaller leaf angles tend to lead to an underestimation in leaf area estimates, whilst greater leaf angles tend to do the contrary and lead to an overestimation [47]. In our study, this may have occurred, particularly for our *C. maculata* results. Although *C. maculata* did not present the highest value for plant area metrics, the number of TLS points was greater than for the other two tree species, which may be explained due to *C. maculata* having pendulous leaves with greater angles of repose.

Additionally, the occlusion effect may play a role in the misrepresentation of the 3D canopy. The high density of plant surfaces may have hindered the retrieval of information from deeper layers

of the canopy. Issues with occlusion are described in studies using TLS data [18,21,46], and the possibility of occlusion occurrence may be reduced with the capturing of multiple scans, which is possible using a mobile terrestrial laser scanner. Even so, occlusion can still happen when collecting data with mobile devices, because internal features cannot be captured from an inside view unless the scanner device is introduced into the canopy, which is still not a parameterised method.

Moreover, ground-based surveys tend not to capture the top part of a tree well, depending on their height. In previous studies, the attenuation of the laser beam and its reflectance has posed a limitation to the use of TLS data [20,42]. In our study, trees were not higher than 2.7 m, which did not limit the representation of the highest region of the canopy. However, the occlusion and attenuation effect may affect the assessment of larger trees, particularly in an outdoor urban environment.

As a limitation, this study did not measure secondary factors that may be causing the difference among species' signatures (e.g., leaf angle). Additionally, using juvenile trees may have underestimated the role of woody parts in laser detection, as larger branches (when older) may be captured more easily by the process. This fact should be considered when scanning mature trees in an urban context, because mature trees tend to present denser canopies, which may intensify the occlusion effect by hindering the laser scanner from penetrating deeper into the canopy [19]. However, as a matter of comparison, the PAI for mature trees in an urban streetscape in Melbourne ranges from 0.6 to 5.2 for *Platanus* × *acerifolia*, 2.1 to 7.6 for *Ulmus procera*, and 1.3 to 3.1 for *Eucalypts scoparia* [48], which is comparable with the measured PAI for the trees in our study (Table 1).

### 4.2. Correlation TLS Data and Rainfall Interception Parameters

*U. procera* presented a greater slope in all regression tests, showing greater water storage capacity when compared to *P. acerifolia* and *C. maculata*, even if the number of scanned points was lower than for the others. This fact indicates that parameters other than plant area must be affecting $C_{min}$, as the number of scanned points is dependent on the presence of a surface to reflect the laser beam with. Higher values of $C_{min}$ are explained by characteristics that promote the canopy water storage capacity, such as greater plant surface area [27], but also leaf and bark hydrophobicity [26,28] and branch and leaf inclination [49], which were not measured during this study.

The significant interaction between species and parameters studied indicates that specific canopy arrangements may explain differences in water storage capacity. In this study, for example, *C. maculata* leaves presented an angle close to 90° in relation to the ground, which may be more easily captured by the terrestrial laser scanning, explaining why *C. maculata* presented the higher values of points per area and volume. The higher inclination angles of leaves and branches create gaps inside the canopy, which allow the penetration of the laser beam from a ground-level scanner. On the other hand, this characteristic also prevents water from sticking to the tree [29] and reduces the water storage in this species.

Additionally, TLS-derived metrics do not take into account other important microscale characteristics of the leaves or bark, such as hairs, waxy cuticles, serrated laminar, and coarse and papery bark layers. Those aspects are important because $C_{min}$ is affected by them, and they were not measured in this study [28,50] and cannot be measured by TLS data of this type.

Branches play an important role in water storage capacity [27]. From the scanned data, the calculated proportion of the number of points shows that 34% of the scanned surface is associated to woody parts for *C. maculata* trees, 28% for *P. acerifolia*, and 55% for *U. procera*. Branch surfaces may store part of the incident rainfall, being responsible for storing up to 40.8% of the intercepted rainfall for *C. maculata*, 11.7% for *P. acerifolia*, and 12.8% for *U. procera* [43]. Branches can also be responsible for draining water out of the plant system via stemflow, which is not counted as intercepted water. Studies have shown that stemflow is an important component of the interception process for some species [51], redirecting up to 10% of incident rainfall to the ground, depending on the rainfall characteristics [49]. However, stemflow was not measured, as this study has focused on water storage capacity and throughfall analyses.

### 4.3. Spatial Redistribution of Throughfall

Under natural rainfall conditions, canopy throughfall redistribution will be influenced by changes in rainfall volume, such that spatial heterogeneity or coefficients of variation decrease as rain volume increases [39,52]. Other abiotic factors that may increase throughfall volume under a tree canopy are extended rainfall duration and higher rainfall intensity [53–55]. However, in the rainfall simulations of this study, these factors were kept constant; therefore, the differences in the spatial distribution of throughfall may be more confidently attributed to specific differences in the canopy characteristics among the tree species.

At a forest scale, a previous study [56] found a significant correlation between throughfall and Light Detection and Ranging derived (LiDAR) metrics in broad-leaved, coniferous, and mixed forests in Denmark. However, the variance in throughfall was best explained by the LiDAR density matrix for annual precipitation rates. The explanatory power of LiDAR-derived data increases as the temporal resolution decreases from monthly to seasonal and annual rainfall. This indicates that for a single rainfall event, the link between point cloud data such as TLS-derived metrics and rainfall will be obfuscated by noise. However, over many events in a season or a year, TLS-derived metrics could provide a prediction of throughfall.

Previous studies have demonstrated a level of uncertainty in regards to throughfall spatial distribution results and correlation with forest attributes because of the high variation of collected data [57,58]. However, as suggested in Ref. [59], an increase in the number of sampling points may improve the quality of collected data, as it may overcome high variation over a small area. In this study, the plot size is 1.87 × 1.98 m and the distance between cups equals 11 cm. Therefore, the total area covered by the vials (6.2 cm$^2$) represents about 5% of the total area, which is greater than in previous works [40].

Statistical analyses have shown a significant correlation between throughfall reduction zones and denser canopy for *U. procera* and *P. acerifolia*. A higher density coincides with areas of lower throughfall because it corresponds to a larger plant surface to store water [60]. However, some trees present hotspots with high values of throughfall under an area of high point density in the middle of the canopy projection area. As can be seen by overlapping the throughfall redistribution with the point density map, the canopy structure may be influencing the redistribution process, as the arrangement of leaves and branches in this region conducts the water that has dropped from other layers to the bottom.

Patterns of distribution are different from one species to another. Visually, maps of throughfall distribution show a varied pattern. Statistically, throughfall maps for *P. acerifolia* trees present higher values for the coefficient of variation compared to the other two species. The significant difference may be influenced by the size of leaves and distribution of branches and leaves in the space. More open canopies, such as is the case with *C. maculata* trees, presented more concentration spots when compared to *P. acerifolia*. However, the volume of water collected on concentration spots under *P. acerifolia* canopy is much higher than the values measured under *C. maculata* canopy. Water seems to be concentrating in the layers over the bottom line, and rainwater has been redirected to these spots. This effect may be explained by taking the approach suggested in Ref. [61]: A tree canopy can be divided into different horizontal layers. Rainfall that has reached a given layer could have either passed through the gap in the canopy or been intercepted by plant surfaces in the next layer. In the case of *P. acerifolia*, with the largest and flattest leaves amongst the three species, the leaves are favouring the interception and formation of a big drop, because all water that drips on the leaf surface takes longer to accumulate before dripping to the next layer. At the same time, these characteristics help with the shelter effect in other areas of the redistribution map.

Previous studies have correlated the spatial redistribution of throughfall underneath tree canopies to the erosive potential of drips reaching the ground surface [35,62]. In natural forests, most of the splash-induced erosion caused by throughfall is hindered by understory vegetation and litter cover [63]. However, in the urban forest, surface litter may not be present and understory vegetation limited to turfgrass only. So, for urban trees planted above either bare soil or sparse understorey vegetation,

especially if the land is sloping, trees that greatly redistribute throughfall may lead to greater erosion and green space damage. Furthermore, increasing the complexity of understorey vegetation beneath green space trees has been demonstrated to increase the infiltration of throughfall and thereby decrease the occurrence of surface water runoff [64].

## 5. Conclusions

Retrieving plant area metrics from TLS data has shown a significant correlation between directly measured and remotely sensed data. However, the regressions shown in this paper displayed important differences in the slope and level of association between TLS data and plant area metrics. In other words, practitioners should exercise caution in making assumptions based on TLS data without considering the species. For example, for a given number of TLS points, a very different plant area metric or $C_{min}$ would be predicted for each of the three tree species in our study.

In addition, the estimate of plant surface metrics may be underrepresented for species with smaller leaves, such as *U. procera*. Yet, the strong relationship between surface area metrics and the number of TLS points confirms the connection between those parameters, encouraging the use of TLS data for these estimated and future studies.

The use of point cloud density as a parameter to estimate tree metrics simplifies the use of TLS data in practical analyses. However, the use of TLS to predict interception parameters may be limited by leaf characteristics, such as leaf angle and size. Leaf size and angle are crucial features to evaluate the quality of TLS to predict interception, because it affects the chance of the leaf to be reached by the laser beam. Other metrics derived from TLS data may be applied and additional studies may help to understand how to apply laser scanning technologies in the context of urban forestry.

Despite throughfall rates and plant density showing a weak correlation for regressions, the reduction of throughfall was associated with the presence of a denser canopy zone. Understanding the patterns of throughfall distribution may help to guide the selection of species, the use of understory vegetation, and type of ground cover used under urban trees. Considering trees' specificities in the selection of species is a key factor that should be considered to achieve a potential rate of rainfall interception when planning to use trees for stormwater management.

**Author Contributions:** Conceptualization, M.D.B., S.J.L., M.N. and M.A.; Data curation, M.D.B.; Formal analysis, M.D.B.; Funding acquisition, S.J.L. and M.A.; Investigation, M.D.B.; Methodology, M.D.B., S.J.L., E.G.P. and M.A.; Project administration, M.D.B. and M.A.; Software, E.G.P.; Supervision, M.A.; Validation, M.D.B.; Visualization, M.D.B.; Writing—original draft, M.D.B.; Writing—review & editing, S.J.L., E.G.P., M.N. and M.A.

**Funding:** This work was supported by the Australian Research Council Discovery Project: "Seeing the good from the trees: remotely sensing the urban forest" (DP 150103135).

**Conflicts of Interest:** The authors declare no conflict of interest.

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
