# Peer review of "Terrestrial Laser Scanning to Predict Canopy Area Metrics, Water Storage Capacity, and Throughfall Redistribution in Small Trees"

_remotesensing, doi:10.3390/rs10121958_

Round 1

Reviewer 1 Report

The work has interest in the methodology of calculation of the correlation between the TLS data (as NP, NPA or NPV) and the absorption of water by the plant (as Cmin) or the interception of water.

However, the number of plants used, only 3 differents, with different leaf density makes correlations are poorly based.

The contribution of the work is the correlation between parameters such as NP and Cmin or the density of water interception, however it extends in explaining the correlation between data obtained by TLS (NP, NPA and NPV). These last are only a verification of previous works.

The leaf area was measured manually and the PSA (Plant Surface Area) and PAI (Plant Area Index) and PAV (Plant Area Volume) were calculated. A correlation between PSA and NP was established; between PAI and NPA and PAV and NPV. These correlations have already been made in previous works and here simply the expected results are checked. But in the end it appears as a significant part of the work discussion.

The water collected in the vials was studied to study the run-off distribution, the throughfall. Subtracting the average rainfall minus the water collected in the vials, the interception distribution is obtained.

Low inclination implies lower water storage capacity, but at the same time a low inclination it implies less detection by the lateral TLS.

In table 3 the correlation for U.procera and C.maculata is very low in the graphs (NP, Cmin) (NPA, Cmin) (NPV, Cmin); being only well correlated the graphs for P.acerifolia. 

However the correlations (NP, PSA) (NPA, PAI) (NPV, PAD) where they are worse correlated is with P.acerifolia. This should not be the case and it is logical that in the plant where there was a good correlation (NPA, PAI) for example there would be good correlation (NPA, Cmin) since the density of points should directly influence the storage capacity.

In line 264 it is indicated that the correlation (NPA, Cmin) is the least significant. However, by concept it should be the most important, since the foliar area should have a maximum impact on the storage capacity of the plant. This is not a good result to justify the work.

The points with the lowest water concentration correspond to the points with the highest vegetative density. But sometimes this is not so because the distribution of the geometry of the tree can cause the water to be collected from other areas of the canopy and tip over the points that correspond to the high density of points.

Figure 3 (Page 8). In 3.1 and 3.2 is not indicated the icon legends. 

The methodology is interesting, but it would take more TLS operations to draw the strongest conclusions.

Author Response

The work has interest in the methodology of calculation of the correlation between the TLS data (as NP, NPA or NPV) and the absorption of water by the plant (as Cmin) or the interception of water.

Firsts of all, we would like to thank the reviewer for all valuable comments on this paper and their interest in the correlation between TLS metrics and the canopy water storage parameter Cmin . This is a central aim of this study, to investigate the ability of remotely sensed data to predict the functional roles of trees in the landscape.

However, the number of plants used, only 3 differents, with different leaf density makes correlations are poorly based.

The authors do not agree with this comment. This study has selected 3 tree species, and 4 trees for each species, totalling 12 studied trees. Those 12 trees had their canopies manipulated to create four canopy density treatments such that there are 48 data points  (16 for each tree species) to investigate the relationships between TLS data, leaf area metrics and Cmin,  which is not a small number and more than adequate to develop correlations from.  

The contribution of the work is the correlation between parameters such as NP and Cmin or the density of water interception, however it extends in explaining the correlation between data obtained by TLS (NP, NPA and NPV). These last are only a verification of previous works.

The leaf area was measured manually and the PSA (Plant Surface Area) and PAI (Plant Area Index) and PAV (Plant Area Volume) were calculated. A correlation between PSA and NP was established; between PAI and NPA and PAV and NPV. These correlations have already been made in previous works and here simply the expected results are checked.

We agree with the reviewer that the correlation between directly measured plant area metrics and other TLS metrics have been reported in previous works. However, previous studies have used different approaches and data. Two paragraphs were added to explain how these correlations were performed differently. These paragraphs read as follows:

“This study proposes a new way to make use of TLS and thereby add value to the work of urban forestry professionals. It investigates the potential of deriving plant area metrics from TLS data clouds, so as to avoid expensive and laborious manual methods of leaf area data collection that require destructive sampling.(…)”(L85)  

“(…) Most of previous approaches have used an algorithm to extract gap fractions from TLS point clouds and from this an approximation of LAI [17], [20], [41], [42]. Two approaches commonly applied have been the 3D voxel-based canopy profiling (VCP) method [41], and a 2D approach which converts “the point cloud data set from Cartesian coordinates to spherical coordinates in order to be similar to hemispherical photography”[20]. The present study differs from cited works by investigating whether the number of scanned points correlates with manually calculated plant area metrics.(…)”(L93)

We believe this is an important base from which to discuss our findings in the following sections of the manuscript.

The water collected in the vials was studied to study the run-off distribution, the throughfall. Subtracting the average rainfall minus the water collected in the vials, the interception distribution is obtained.

Low inclination implies lower water storage capacity, but at the same time a low inclination it implies less detection by the lateral TLS.

We agree that with standard lateral TLS a low inclination would imply less detection. However, the mobile laser scanner (Zebedee) used allows us to walk around the tree with a constantly and randomly changing location (head) for scanning, such that the lateral angles vary constantly, decreasing significantly any misdetection effect.

In table 3 the correlation for U.procera and C.maculata is very low in the graphs (NP, Cmin) (NPA, Cmin) (NPV, Cmin); being only well correlated the graphs for P.acerifolia.

However the correlations (NP, PSA) (NPA, PAI) (NPV, PAD) where they are worse correlated is with P.acerifolia. This should not be the case and it is logical that in the plant where there was a good correlation (NPA, PAI) for example there would be good correlation (NPA, Cmin) since the density of points should directly influence the storage capacity.

In line 264 it is indicated that the correlation (NPA, Cmin) is the least significant. However, by concept, it should be the most important since the foliar area should have a maximum impact on the storage capacity of the plant. This is not a good result to justify the work.

We agree that the correlation between the number of TLS points and Cmin is best for P. acerifolia, and not as good for U. procera and C. maculata. However, these correlations are still statistically significant for U. procera and C. maculata.

We agree if we only consider the density of points as a representation of plant surface area that this would be an indicator of a larger area for potential water storage. However, other surface characteristics that are not captured by the TLS influence to increase storage capacity. For example, imagine we have two cubes with a similar surface area and one is made of stainless steel and the other is made of cotton. We can assume that the number of scanned points would be similar. On the other hand, the storage capacity of the steel cube would be much smaller compared to the cotton cube.  To clarify this discussion, we have added a paragraph that reads as follows:

“Additionally, TLS derived metrics do not take into account other important micro-scale characteristics of the leaves or bark, such as hairs, waxy cuticles, serrated laminar, coarse and papery bark layers. Those aspects are important because Cmin is affected by them and those were not measured in this study [28], [50], and cannot be measured by TLS data of this type (L 411)”.

Additionally, it is important to acknowledge that the strength of correlations is not the only important result. Perhaps more importantly, these regressions also show important differences between the three tree species in how TLS data could be used to predict either plant area metrics or ecosystem functions such as Cmin. For example, for a given number of TLS points (e.g. 20,000), a very different plant area metric or Cmin would be predicted for each of these three tree species. A section was added to the conclusions to make this statement clear. The paragraph now read as follows:

“Retrieving plant area metrics from TLS data has shown a significant correlation between directly measured and remotely sensed data. However, the regressions shown in this paper displayed important differences in the slope and level of association between TLS data and plant area metrics. In other words, practitioners should exercise caution in making assumptions based on TLS data without considering the species. For example, for a given number of TLS points, a very different plant area metric or Cmin would be predicted for each of the three tree species in our study. (L 474)”

Line 264 does not show a correlation between NPA and Cmin, so we hope we have addressed any potential confusion with the previous response.

The points with the lowest water concentration correspond to the points with the highest vegetative density. But sometimes this is not so because the distribution of the geometry of the tree can cause the water to be collected from other areas of the canopy and tip over the points that correspond to the high density of points.

We agree and acknowledge that this topic was covered in our discussion section.

Figure 3 (Page 8). In 3.1 and 3.2 is not indicated the icon legends. 

Figure 3 was adjusted.

The methodology is interesting, but it would take more TLS operations to draw the strongest conclusions.

We agree with the reviewer. 

Reviewer 2 Report

Manuscript #384388 uses terrestrial LiDAR scanning to estimate canopy structures for common street trees and relate them to water storage/throughfall patterns under simulated rainfall. I agree with the authors that this type of work is needed. But, I believe several major revisions must be accomplished before the manuscript can be published. The major revisions are related to the hydrometeorological methods. The LiDAR data collection and processing is clear and correct, to my knowledge.

Section 2.1: Can the authors please provide a comparison of their study trees’ structural traits to those of actual trees of the same species planted in streetscapes throughout Melbourne? If these potted trees are differently structured compared to trees in Melbourne streetscapes, it weakens the study justification and real-world applicability.

Section 2.2: What was the purpose of the leaf removal manipulation? Are there situations where street trees would become defoliated in such a manner, losing “every other leaf on a branch,” then losing every other leaf again and again? This doesn’t seem, to me, to represent any defoliation processes in nature.

Section 2.3: I understand why trees were “top-trimmed to… 2.7 m” for the rainfall simulation. But, is this really representative of street tree canopies in Melbourne? This, and other, manipulations may severely diminish the value of the authors’ results to urban forest-water management.

Section 2.4: Why do the authors choose *minimum* water storage capacity for relating to TLS-derived metrics? Hydrologic models rely on *maximum* water storage capacity and the authors estimated max water storage capacity from the continuous weight monitoring. The term (minimum water storage capacity) is also made hazier by its being typically used by the precipitation partitioning field in reference to a specific way of estimating “maximum water storage capacity” per the Leyton method (Klaassen et al., 1998, 10.1016/S0022-1694(98)00200-5).

Section 2.5: The throughfall pattern measurements were thorough! Unfortunately, the gridded design has been shown to less accurately reveal throughfall patterns (Voss et al., 2016, 10.1016/j.jhydrol.2016.06.042). Studies on throughfall pattern estimation and analysis have found that “transect” sampling strategies are superior to grid-based designs. Rather than re-running rainfall simulations, I recommend providing a brief discussion of the impact of sampling strategy on throughfall pattern analysis and what it may mean for the results of this study.

More minor comments on the writing/referencing:

I suggest objective 1 should be removed from the manuscript and placed in a supplement. As admitted by the authors, many studies show that TLS-derived metrics compare well with (and are in many cases more accurate than) standard measurement methods.

L82: Research linking throughfall to the biodiversity and function of soil microbes post-dates Levia and Frost (2003) – Rosier et al. (2015, 10.1002/eco.1595) and Moore et al. (2016, 10.1016/j.soilbio.2016.05.016). Moreover, the authors do not measure stemflow, so why reference this stemflow review article?

L82 (re: roots): only Ford and Deans (1978, 10.2307/2402786) have ever observationally linked throughfall patterns to fine root patterns in forests.

L83-84: Multiple publications by Zimmermann discuss how canopy traits relate to throughfall patterns (maybe start here: Zimmermann et al., 2007, 10.1016/j.jhydrol.2007.06.012). 

Author Response

Reviewer 2

Manuscript #384388 uses terrestrial LiDAR scanning to estimate canopy structures for common street trees and relate them to water storage/throughfall patterns under simulated rainfall. I agree with the authors that this type of work is needed. But, I believe several major revisions must be accomplished before the manuscript can be published. The major revisions are related to the hydrometeorological methods. The LiDAR data collection and processing is clear and correct, to my knowledge.

 First of all, we would like to thank the reviewer for all valuable comments on this paper.

Section 2.1: Can the authors please provide a comparison of their study trees’ structural traits to those of actual trees of the same species planted in streetscapes throughout Melbourne? If these potted trees are differently structured compared to trees in Melbourne streetscapes, it weakens the study justification and real-world applicability.

The structural traits are indeed different between the trees used in this study and streetscape planted trees. However, this study provides a good first step to measure interception dynamics, leaf area metrics and to collect TLS point cloud data in a detailed and replicated manner. We have added a section in the manuscript to clarify this equivalence, and it reads as follows:

“However, as a matter of comparison, the PAI for mature trees in an urban streetscape in Melbourne ranges from 0.6 to 5.2 for Platanus x acerifolia, 2.1 to 7.6 for Ulmus procera, and 1.3 to 3.1 for Eucalypts scoparia, which is comparable with measured PAI for trees in our study (Table 1) (L 393)”.

Although differences exist, for the 3 studied species, plant area index does not differ between juvenile trees in pot and mature trees in the streetscape. Differences are observed for U. procera bark, which becomes rougher in mature ages. Even so, U. procera presents a rougher bark compared to the other 2 species when in its juvenile stage.

We believe that the applicability of this study is still relevant, as data may help urban foresters to predict interception parameters using derived TLS metrics. However, we acknowledge that the title of this paper may have suggested to some broader assumptions and based on this comment we decided to retitle this article as: “Terrestrial Laser Scanning to Predict Canopy Area Metrics, Water Storage Capacity and Throughfall Redistribution of Small Trees”.

Section 2.2: What was the purpose of the leaf removal manipulation? Are there situations where street trees would become defoliated in such a manner, losing “every other leaf on a branch,” then losing every other leaf again and again? This doesn’t seem, to me, to represent any defoliation processes in nature.

We agree that the purpose of the leaf manipulation needed to be better explained. A sentence was added to line 122, and read as follows:

“The leaf density was manipulated with the purpose of simulating differences in storage capacity from a canopy in good to poor health. (…)” (L132)

Additionally, we do not believe that this reduction needs to replicate any process in nature, as it aimed to simply lead to a range of canopy leaf densities in the same tree architecture. In other words, this manipulation aims that only the number of leaves changes, as nothing else change, as there are many events whereby tree canopies can lose leaves, such as herbivory,  hail storms, heatwaves, severe drought, etc.

Section 2.3: I understand why trees were “top-trimmed to… 2.7 m” for the rainfall simulation. But, is this really representative of street tree canopies in Melbourne? This, and other, manipulations may severely diminish the value of the authors’ results to urban forest-water management.

Many street trees in Melbourne and other cities that retain overhead power lines are commonly lopped to maintain a clearance zone between the top of the tree and the power lines.

Section 2.4: Why do the authors choose *minimum* water storage capacity for relating to TLS-derived metrics? Hydrologic models rely on *maximum* water storage capacity and the authors estimated max water storage capacity from the continuous weight monitoring. The term (minimum water storage capacity) is also made hazier by its being typically used by the precipitation partitioning field in reference to a specific way of estimating “maximum water storage capacity” per the Leyton method (Klaassen et al., 1998, 10.1016/S0022-1694(98)00200-5).

Previous studies have considered Cmin (or storage capacity) as the actual interception loss as that is the amount of rain that stick to the surface after rain ceases and never reach the ground (Xiao and McPherson, 2016 10.2134/jeq2015.02.0092; Li et al., 2016, 10.1016/j.agrformet.2015.11.017; Dijk et al., 2015, 10.1016/j.agrformet.2015.09.006). We considered this data to be more useful to planners when using trees as a tool for reducing stormwater runoff, as this is the volume of water that is actually intercepted.
Additionally, in our previous work (Baptista et al 2018,
10.1002/hyp.13255) we have found a good correlation between canopy metrics and Cmin.

Section 2.5: The throughfall pattern measurements were thorough! Unfortunately, the gridded design has been shown to less accurately reveal throughfall patterns (Voss et al., 2016, 10.1016/j.jhydrol.2016.06.042). Studies on throughfall pattern estimation and analysis have found that “transect” sampling strategies are superior to grid-based designs. Rather than re-running rainfall simulations, I recommend providing a brief discussion of the impact of sampling strategy on throughfall pattern analysis and what it may mean for the results of this study.

Voss et al. (2016) recommend considering an extent that exceeds several times the “true” range, a minimum of 150 samples, a transect design covering several small distances (0.1 – 1.5 m). They do not recommend a grid-based design to collect throughfall because it may not capture the variance at small lags. However, those tests were made on a larger scale, considering plots of 25m, 50m and 100 m length. According to Voss et al. (2016), the problem is not in the grid-based design by itself but in the resolution of data collection. We have added a paragraph in the discussion that reads as follows:

“In this study, the plot size is 1.87 x 1.98 m and the distance between cups equals 11 cm. Therefore, the total area by the vials (6.2 cm2) represents about 5% of the total area, which is greater than previous works(Zimmermann & Zimmermann 2014).” (L 438)

More minor comments on the writing/referencing:

I suggest objective 1 should be removed from the manuscript and placed in a supplement. As admitted by the authors, many studies show that TLS-derived metrics compare well with (and are in many cases more accurate than) standard measurement methods.

We agree with the reviewer that the correlation between directly measured plant area metrics and TLS data have been reported in previous works. However, these other studies have used different approaches. In this study, we have correlated the number of scanned point with directly measured plant surface for three different species, aiming to understand the interaction between the number of points and species.  A paragraph was added to explain that these correlations were performed differently from previous ones. Additionally, we believe this objective is important to guide our findings in the next sections and would not agree to remove it.

“(…) Most of the previous approaches have used an algorithm to extract gap fractions from TLS point clouds and from this an approximation of LAI [17], [20], [41], [42]. Two approaches commonly applied have been the 3D voxel-based canopy profiling (VCP) method [41], and a 2D approach which converts “the point cloud data set from Cartesian coordinates to spherical coordinates in order to be similar to hemispherical photography”[20]. The present study differs from cited works by investigating whether the number of scanned points correlates with manually calculated plant area metrics for three different species.(…)”(L93)

L82: Research linking throughfall to the biodiversity and function of soil microbes post-dates Levia and Frost (2003) – Rosier et al. (2015, 10.1002/eco.1595) and Moore et al. (2016, 10.1016/j.soilbio.2016.05.016). Moreover, the authors do not measure stemflow, so why reference this stemflow review article?

We acknowledge that stemflow was not measured in this study and changed the reference, Levia and Frost (2003), with the references suggested. 

L82 (re: roots): only Ford and Deans (1978, 10.2307/2402786) have ever observationally linked throughfall patterns to fine root patterns in forests.

We agree that the reference, Levia and Frost (2003), is not relevant for this study. The reference was substituted by the suggested studies.

L83-84: Multiple publications by Zimmermann discuss how canopy traits relate to throughfall patterns (maybe start here: Zimmermann et al., 2007, 10.1016/j.jhydrol.2007.06.012).

We appreciate the reviewer’s suggestions and used the reference in our discussion.

Reviewer 3 Report

Dear Autors,

The presented experiments and comparisons are a very valuable addition to knowledge in this field. The present study describes the use of TLS data as a tool for measuring tree parameters and hydrological value such as canopy storage capacity. 

I really like the idea of reducing the surface of the crown.

I suggest minor corrections:

The abstract corresponds very well to the content of the article.

Methodology carefully described.

4 trees within 1 species is the minimum amount for the correct statistic.. 

Not only urban trees - I recommend extending the audience to:

trees in the urban and natural forest.

Of course, it can be noted that in the forest trees grow in close contact and there would be a necessity to scan the whole layer of crowns. Nevertheless, writing that only urban trees have an intercept function is incomplete.

Quotation 23 - please replace with newer ones:

 Variability in the Wettability and Water Storage Capacity of Common Oak Leaves (Quercus robur L.); A Klamerus-Iwan, W Witek. Water 10 (6), 695

or Canopy storage capacity and wettability of leaves and needles: The effect of water temperature changes; A Klamerus-Iwan, E Błońska. Journal of Hydrology 559,

Author Contributions, Acknowledgments; Conflicts of Interest = require improvement and adaptation to the requirements of the journal

best regards

Author Response

Reviewer 3

Dear Authors,

The presented experiments and comparisons are a very valuable addition to knowledge in this field. The present study describes the use of TLS data as a tool for measuring tree parameters and hydrological value such as canopy storage capacity. 

I really like the idea of reducing the surface of the crown.

First of all, we would like to thank the reviewer for all valuable comments on this paper.

We appreciate that the reviewer acknowledged the importance of this study and found it appropriate for the journal scope.

I suggest minor corrections:

The abstract corresponds very well to the content of the article.

Methodology carefully described.

4 trees within 1 species is the minimum amount for the correct statistic..

We agree that the number of objects is important for statistical purpose. Although we have only 4 trees for each species, we manipulated the canopy of these four trees four times which creates 16 statistical units of data per species. We regard this as a sufficient with which to statistically investigate regression relationships between measured parameters.

Not only urban trees - I recommend extending the audience to:

trees in the urban and natural forest.

Of course, it can be noted that in the forest trees grow in close contact and there would be a necessity to scan the whole layer of crowns. Nevertheless, writing that only urban trees have an intercept function is incomplete.

We agree with the reviewer that not only urban trees have an interception function. However, because this study is applied to small isolated trees that were looped, the authors would prefer not make assumptions by extending the findings to trees grown together in a stand so that they form a near continuous canopy, such as in natural or managed forests.

Based on this comment, and another reviewer’s comment, we decided to retitle this article as: “Terrestrial Laser Scanning to Predict Canopy Area Metrics, Water Storage Capacity and Throughfall Redistribution in Small Trees”.

 Quotation 23 - please replace with newer ones:

 Variability in the Wettability and Water Storage Capacity of Common Oak Leaves (Quercus robur L.); A Klamerus-Iwan, W Witek. Water 10 (6), 695

or Canopy storage capacity and wettability of leaves and needles: The effect of water temperature changes; A Klamerus-Iwan, E Błońska. Journal of Hydrology 559,

We accepted the suggestion and have replaced the references.

Author Contributions, Acknowledgments; Conflicts of Interest = require improvement and adaptation to the requirements of the journal

The suggestions were adapted to the journal requirement. 

Round 2

Reviewer 1 Report

Getting Plant Index from LiDAR impacted points is very interesting. It could allow a direct method to obtain then. This detail increase the value of the work. However, individual kind-level regresion is required, how is explained.

Author Response

Reviewer 1

Getting Plant Index from LiDAR impacted points is very interesting. It could allow a direct method to obtain then. This detail increase the value of the work. However, individual kind-level regression is required, how is explained.

We wish to thank the reviewer for this suggestion. We do indeed agree that we are trying to link Lidar returns to the plant area index. We see one of the main contributions of our work to add to the existing understanding of how this relationship depends on species.

Reviewer 2 Report

The revised manuscript has addressed all my previous major concerns. I request minor revisions prior to publication:

Lines 67-74: Water storage capacity of bare canopies typically represent, maximum, a few mm of rainfall (and, <1 mm in the authors’ study). The argument (and study justification) that canopy rainfall interception can be an important factor in stormwater control would be strengthened if the authors mentioned that these few mm of canopy water storage capacity fuel evaporation which, together, can significantly reduce rainfall receipt at the surface for even large storms.

Line 69, 71, 100: please revise “storage capacity” to “water storage capacity” throughout, or consistently use Cmin, after its introduction, throughout.

Lines 157-160: The authors’ point, that many urban trees are top-trimmed to maintain clearances between the tree canopy and urban structures (like power lines), should be included here. One sentence would let readers know that top-trimming does not diminish the real-world value of the authors' rainfall simulation results.

Lines 388-391: I would place the comparison between selected trees and those in the urban streetscape up front, in the methods section 2.1.

Line 417: Please remove the space in “stem_flow” throughout, as this term is typically written without the space, “stemflow.”

Section 4.3: I recognize that this study focuses on throughfall spatial patterns at the individual canopy scale, but perhaps this discussion section would benefit from a brief comparison with results from a study of larger-scale throughfall patterns and airborne lidar-derived canopy density metrics by Schumacher and Christiansen (2015: https://doi.org/10.1016/j.agrformet.2014.12.007).

Author Response

The revised manuscript has addressed all my previous major concerns. I request minor revisions prior to publication:

We are glad to hear that previous major concerns were addressed. We would like to thank reviewer 2 for their valuable comments.

Lines 67-74: Water storage capacity of bare canopies typically represent, maximum, a few mm of rainfall (and, <1 mm in the authors’ study). The argument (and study justification) that canopy rainfall interception can be an important factor in stormwater control would be strengthened if the authors mentioned that these few mm of canopy water storage capacity fuel evaporation which, together, can significantly reduce rainfall receipt at the surface for even large storms.

We agree with this suggestion and added a sentence to the paragraph that reads as follows:

 “In the case of water storage capacity, plant surface area, roughness, and angle of inclination influence the volume of water that can be stored during and after rainfall ceases. Further, woody parts of the tree intercept rainfall and drops from leaves providing additional area for evaporation and drying. Fundamentally (…)”

Line 69, 71, 100: please revise “storage capacity” to “water storage capacity” throughout, or consistently use Cmin, after its introduction, throughout.

We agree with the reviewer. The term “storage capacity” was replaced with “water storage capacity” in the indicated lines and throughout the manuscript, and “Cmin” was used when appropriate.

 Lines 157-160: The authors’ point, that many urban trees are top-trimmed to maintain clearances between the tree canopy and urban structures (like power lines), should be included here. One sentence would let readers know that top-trimming does not diminish the real-world value of the authors' rainfall simulation results.

We agree with this suggestion and added a sentence to the paragraph that reads as follows:

“All trees were top-trimmed to ensure they did not exceed a height of 2.7 m so that rainfall distribution was uniform, once it intercepted the top of a tree canopy. Many street trees in Melbourne and other cities that retain overhead power lines are commonly lopped to maintain a clearance zone between the top of the tree and the power lines. Our experimental set up, therefore, replicates real-world conditions. (…)”

Lines 388-391: I would place the comparison between selected trees and those in the urban streetscape up front, in the methods section 2.1.

We kept the comparison between selected trees and those in the urban streetscape in the discussion section because it helps to explain the limitations of the use of small trees in this study.

 Line 417: Please remove the space in “stem_flow” throughout, as this term is typically written without the space, “stemflow.”

We agree with the reviewer. All spaces were removed throughout the manuscript.

 Section 4.3: I recognize that this study focuses on throughfall spatial patterns at the individual canopy scale, but perhaps this discussion section would benefit from a brief comparison with results from a study of larger-scale throughfall patterns and airborne lidar-derived canopy density metrics by Schumacher and Christiansen (2015: https://doi.org/10.1016/j.agrformet.2014.12.007).

We accepted the suggestion and a paragraph was added to the discussion section and reads as follows:

“At a forest scale, Schumacher and Christiansen (2015) found a significant correlation between throughfall and LiDAR derived metrics in broad-leaved, coniferous and mixed forests in Denmark. However, the variance in TF was best explained by the LiDAR density matrix for annual precipitation rates. The explanatory power of LiDAR derived data increases as the temporal resolution decreases from monthly to seasonal and annual rainfall. This indicates that for a single rainfall event the link between point cloud data such as TLS derived metrics and rainfall will be obfuscated by noise. However, over many events in a season or a year, TLS derived metrics could provide a prediction of throughfall. “